# Effects of Sodium Montmorillonite on the Preparation and Properties of Cellulose Aerogels

**DOI:** 10.3390/polym11030415

**Published:** 2019-03-04

**Authors:** Lin-Yu Long, Fen-Fen Li, Yun-Xuan Weng, Yu-Zhong Wang

**Affiliations:** 1School of Materials and Mechanical Engineering, Beijing Technology& Business University, Beijing 100048, China; 15652591908@163.com (L.-Y.L.); 13264056919@163.com (F.-F.L.); 2Beijing Key Laboratory of Quality Evaluation Technology for Hygiene and Safety of Plastics, Beijing Technology and Business University, Beijing 100048, China; 3Center for Degradable and Flame-Retardant Polymeric Materials, College of Chemistry, Sichuan University, Chengdu 610064, China

**Keywords:** cellulose, montmorillonite, composite aerogel

## Abstract

In this study, first, a green and efficient NaOH/urea aqueous solution system was used to dissolve cellulose. Second, the resulting solution was mixed with sodium montmorillonite. Third, a cellulose/montmorillonite aerogel with a three-dimensional porous structure was prepared via a sol-gel process, solvent exchange and freeze-drying. The viscoelastic analysis results showed that the addition of montmorillonite accelerated the sol-gel process in the cellulose solution. During this process, montmorillonite adhered to the cellulose substrate surface via hydrogen bonding and then became embedded in the pore structure of the cellulose aerogel. As a result, the pore diameter of the aerogel decreased and the specific surface area of the aerogel increased. Furthermore, the addition of montmorillonite increased the compressive modulus and density of the cellulose aerogel and reduced volume shrinkage during the preparation process. In addition, the oil/water adsorption capacities of cellulose aerogels and cellulose/montmorillon aerogels were investigated.

## 1. Introduction

Today, because the depletion of petroleum resources and serious environmental pollution continue to increase, the preparation of cellulose-based functional materials has attracted increasing attention from scientists and society [1,2]. Abundant raw materials with a high natural polymer-cellulose content have been a particular focus of recent studies. Among previously developed cellulose-based functional materials, cellulose aerogels have become a research hotspot because they can be fabricated at a low cost as well as exhibit excellent biocompatibility and cellulose biodegradability, green reproducibility, a low density, high porosity and a large specific surface area [3,4,5,6,7,8].

The general method used for preparing cellulose aerogels involves cellulose dissolution/dispersion followed by a sol-gel process and freeze-drying or supercritical drying [8]. In a previous study by Geng, a NaOH/urea aqueous solution was used as a solvent for cellulose, *N*,*N*′-methylenebisacrylamide was used as a crosslinking agent, and freeze-drying was used to prepare cellulose aerogels. The obtained aerogel possessed a three-dimensional porous structure with large pores (20–600 μm), high porosity (90.30–99.02%) and a low density (0.0820–0.0083 g/cm^−3^) and was able to adsorb methylene blue (115 mg/g) and Cu/(Ⅱ) (85 mg/g) [9]. In addition, Duong et al. applied nylon as an inner layer, cellulose aerogel as a middle layer and neoprene as an outer layer using a zigzag stitching method to fabricate a three-layer thermal sheath with excellent thermal insulation properties. The water bottle containing ice water was wrapped with this fabricated heated jacket and still retained a temperature at 0.1 °C after 4 h [10]. However, because of the polyhydroxy structure of and refractory properties of cellulose, pure cellulose aerogel materials have difficulty meeting the functionality, durability and uniformity requirements need for practical applications.

To produce high-performance cellulose composite aerogels, researchers have introduced inorganic or organic materials by physical or chemical methods. Li et al. fabricated cellulose composite aerogels with excellent and highly selective oil/water adsorption capacity by immersing NaIO_4_-oxidized cellulose hydrogels in a chitosan solution and subsequently freeze-drying and modifying the hydrogels with hydrophobic cold plasma treatment [11]. Ren et al. prepared cellulose/graphene oxide composite aerogels with a three-dimensional thin-walled pore structural network, a large specific surface area, and an excellent dye elimination effect (99.0%) by mixing a graphene oxide suspension with a cellulose solution [12]. Additionally, Ge et al. used boric acid as a crosslinking agent to fabricate cellulose/graphene oxide composite aerogels with low thermal conductivity (0.0417 W/m K) [13]. Liu et al. synthesized cellulose nanocrystal-grafted-acrylic acid aerogels with a high swelling ratio (495:1) and excellent methylene blue adsorption capacity (>400 mg/g) using the hydrothermal method and freeze-drying [14]. Fan et al. prepared cellulose nanofiber/AlOOH composite aerogels with flame-retardant and thermal insulation properties (thermal conductivity: 0.0385 W/m K) using the hydrothermal method [15]. Hwang et al. prepared composite aerogel beads by mixing a cellulose solution with Chlamydomonas and Nostoccus cells at concentration of 0.1, 0.3 and 0.5% (*w*/*w*), respectively, and the highest Cd^2+^ removal rate achieved was 90.3% [16].

Sodium-based montmorillonite possesses excellent swelling properties, cation exchangeability, dispersibility in aqueous medium, viscosity, lubricity, and thermal stability as well as high hot and wet compressive strength and compressive modulus [17,18]. It is one of the most ideal cellulose aerogel reinforcing materials. In this paper, a cellulose/montmorillonite composite aerogel with a three-dimensional porous structure was prepared by dissolving cellulose in a green and efficient NaOH/urea aqueous solution system and subsequently mixing the resulting solution with sodium montmorillonite and subjecting the resulting suspension to a sol-gel process, solvent replacement and freeze-drying. For the first time, the effects of montmorillonite on the preparation and properties of cellulose aerogels were studied using a rotational rheometer, scanning electron microscope (SEM), dynamic thermomechanical instrument and specific surface area analyzer. In addition, the liquid adsorption properties of the composite aerogels were examined.

## 2. Materials and Methods

### 2.1. Chemicals and Reagents

Microcrystalline cellulose (65-µm particle size, purity > 95%), NaOH (99%) and urea (99%) were purchased from Macklin Chemical Co. Ltd. (Shanghai, China). Sodium montmorillonite with a mesh of 300 was provided by Zhejiang Fenghong New Material Co., Ltd. (Huzhou, China). All reagents and solvents were of laboratory grade and were used without further purification.

### 2.2. Fabrication of the Cellulose/Montmorillonite Composite Aerogel

The surface morphology and pore structure of the aerogels primarily depend on the drying method used in their fabrication [19,20]. The preparation process of cellulose/montmorillonite composite aerogel used in this study is shown in Figure 1. A simple, environmentally friendly, vacuum freeze drying method was chosen.

First, 5 g cellulose was dissolved in 100 g of pre-cooled NaOH:urea:water (7:12:81, wt %) solution. Second, the solution was centrifuged for ten minutes (4000 revolutions/min) to obtain a uniform and transparent cellulose solution. A 3 g aliquot of montmorillonite was added to the cellulose solution, which was stirred vigorously for 2 h to produce a homogenous cellulose/montmorillonite mixture. Third, the mixture was transferred to a mold (15.5 mm × 17.5 mm), which was incubated for 4 h at 30 °C to obtain a cellulose/montmorillonite hydrogel. Fourth, the hydrogel was immersed in deionized water to remove NaOH and urea. Water was exchanged every 6 h until the last cleaning solution tested neutral. Finally, the obtained hydrogels were freeze-dried suing a lyophilizer (LYO-50FS; Beijing Kaiyuan Yongsheng Freeze Technology Co., Ltd., Beijing, China) for 72 h (−55 °C, <1 Pa) to obtain aerogels.

### 2.3. Characterization

Rheological measurements of the cellulose and montmorillonite mixture were performed at 25 °C on a MCR 200 rheometer (Anton Paar Trading Co., Ltd., Shanghai, China) at a constant shear frequency (1 Hz). The cross-sectional surface structure of the aerogel was observed using an XL field emission SEM (Phenom, Holland) with an energy dispersive spectrometer (EDS) after the aerogel sample was frozen using liquid nitrogen, crushed and sprayed with gold. The mechanical properties of the aerogel were determined by a dynamic thermomechanical analyzer (Hitachi Instruments Co., Ltd., Shanghai, China) at a minimum compression force of 10 mN and a force amplitude default value of 200 mN. The samples were cylindrical with a diameter of 10 mm and a height of 10 mm. Three replicates were tested for each sample. Nitrogen adsorption experiment of aerogels were conducted using an automatic specific surface area and microporous physical adsorption instrument (MicrotracBEL Corp, MicrotracBEL). The density of an aerogel was obtained by measuring the weight and volume of the aerogel. Five commonly used liquids (water, lubricating oil 220# and 48#, ethanol and 1-butanol) were selected for the adsorption experiments. The dimensions of the dry aerogels were 10 mm (diameter) by 8 mm (height). Aerogels were weighed and then immersed in liquid (100 mL) for 10 min. Then, the aerogel was removed from the liquid, allowed to drip dry above the liquid for 30 s to remove residual liquid on the surface, and weighed again. The adsorption capacity of the aerogel was calculated using the following formula:C_a_ = (m_t_ − m_0_)/m_0_(1)
where C_a_ (g/g) represents the adsorption capacity of the aerogel at 10 min and m_t_ (g) and m_0_ (g) represent the masses of the aerogel after or before adsorption, respectively.

## 3. Results and Discussion

### 3.1. Effect of Montmorillonite on Gelation Behavior of Cellulose Solution

The results of the viscoelastic behavior of a cellulose solution and cellulose/sodium montmorillonite suspension at constant temperature (30 °C) and frequency (1 Hz) are shown in Figure 2.

Figure 2 shows that the gelation time of the cellulose/montmorillonite suspension was <300 s, while the gel time of the cellulose solution was >800 s. Figure 1 shows that the cellulose solution and the cellulose/sodium montmorillonite suspension display similar rheological behavior, and their storage modulus (G’) and loss modulus (G’’) exhibit obvious time dependence. Initially, G’ < G’’, and the samples were both viscous liquids. As time increased, G’ increased faster than G’’ and eventually intersected G’’ and then exceeded G’’. The intersection of G’ and G’’ is the gel point, the time point at which the cellulose solution/suspension changes from the liquid phase to the gel phase [9,21,22].

Cellulose solutions/suspension form cellulose gels under the physical action of van der Waals force, hydrogen bonding, hydrophobic or electron association and chain entanglement. In addition, the gelation rate generally depends on the cellulose concentration and solution/suspension temperature, among other factors [23,24]. The addition of a chemical crosslinking agent or materials possessing oxygen-containing groups generally accelerates gelation [9,21,25]. In addition, the initial G’ and G’’ of the cellulose/montmorillonite suspension are larger than those of the cellulose solution, indicating stronger intermolecular forces and higher cellulose chain entanglement occur in the cellulose/montmorillonite suspension because of the addition of montmorillonite. We speculate that the formation of hydrogen bonds between the oxygen-containing groups in the montmorillonite and –OH groups on cellulose molecular chains accelerates the gelation process of the cellulose solution [26,27].

### 3.2. Effect of Montmorillonite on Surface Morphology of Cellulose Aerogel

SEM images of the cross-sections of the cellulose aerogels with different montmorillonite ratio are shown in Figure 3. The EDS spectrum of a cellulose-clay aerogel is shown in Figure 4.

Figure 3 shows that both the cellulose and the cellulose/montmorillonite aerogel were composed of interconnected sheet-like skeletons that formed three-dimensional networks with a large pore size. However, several cracks formed because, during the freeze-drying process, the gel was frozen at a temperature below the freezing point of the liquid medium (which was usually water). Thus, the process relies primarily on sublimation to eliminate the liquid [13], and ice crystals growth and the high interfacial tension of the water caused cracks and the formation of sheet network structures in the aerogel material [19]. Simultaneously, a large amount of particulate matter was attached to the pore wall of the cellulose/montmorillonite aerogel, and the EDS spectrum (Figure 4) showed that the elemental composition consisted of Na, Mg, Al and Si. Furthermore, the average pore size decreased from 87.3–370 μm range for pure cellulose aerogel to the 295 nm–6.75 µm range for 3-wt % cellulose/montmorillonite aerogel, which was similar to the aerogel pore size reported by Zhou et al. [28]. The results of the N_2_ adsorption experiment showed that the specific surface area increased from 7.79 m^2^/g to 23.18 m^2^/g. In total, the analysis results indicated montmorillonite adhered to the surface of the cellulose substrate by hydrogen bonding and was embedded in the pore structure of the cellulose aerogel. As a result, the pore diameter of the aerogel decreased and the specific surface area of the aerogel increased.

### 3.3. Effect of Montmorillonite on the Compressive Modulus of Cellulose Aerogel

To study the effect of the content of montmorillonite on the mechanical properties of cellulose aerogel, we added montmorillonite with a mass fraction of 0, 1, 2 and 3 wt % (relative to the quality of the cellulose solvent) to the cellulose solution. The compressive modulus and density of cellulose aerogel, cellulose/montmorillonite composite aerogel are shown in Figure 5.

Figure 5 shows that the density and compressive modulus of the cellulose aerogel were proportional to the montmorillonite content. The density of cellulose aerogel increases from 0.107 g/cm^3^ to 0.174 g/cm^3^, which was similar to the findings of Sescousse et al., Duchemin et al. and Li et al. [29,30,31]. The aerogel volume shrinkage of aerogel decreased from 50.73% to 45.14%, indicating that cellulose/montmorillonite hydrogels possess a more stable three-dimensional structure and strong internal intermolecular forces.

The mechanical properties of cellulose aerogels are one of the important for practical applications and are generally related to the cellulose concentration, crosslinker concentration and density, among other factors [32,33]. This study showed that the addition of montmorillonite increased the compressive modulus of the cellulose aerogel. Two reasons could explain this phenomenon: (1) The addition of montmorillonite increased the density of the cellulose aerogel, thereby increasing the compressive modulus of the aerogel, (2) The formation of hydrogen bonds between the montmorillonite and cellulose molecules enhanced the structural stability of the aerogel.

### 3.4. Effect of Montmorillonite on Adsorption Properties of Cellulose Aerogels

Figure 6 shows the adsorption capacities of five liquids on cellulose/montmorillonite aerogels with montmorillonite ratios of 0, 1, 2 and 3 wt %. 

Figure 6 shows that the adsorption capacity range of cellulose/montmorillonite aerogel for water, oil and organic solvents was between 3 g/g and 7 g/g. The adsorption capacity of the aerogel was comparable to that of commercial adsorbent polypropylene. In addition, because of the hydrophilicity of cellulose aerogels, the adsorption capacity of water was greater than the adsorption capacity of oil in this study. 

The oil adsorption properties of cellulose aerogels are related to the density, viscosity and surface tension of oily liquids and also depend on capillary effects, van der Waals forces and hydrophobic interactions as well as the density and morphological characteristics (e.g., surface wettability, total pore volume and pore structure) of cellulose aerogels [34,35,36,37]. The low microporous and mesoporous ratio of cellulose aerogel caused by the freeze-drying process was the primary reason for the low oil/water adsorption capacity observed in this study. Our future research will focus on the preparation of low-density, high-porosity cellulose aerogels with excellent adsorption capacity. 

## 4. Conclusions

Cellulose/montmorillonite composite aerogel with a three-dimensional porous structure was fabricated by dissolving cellulose in a green and efficient NaOH/urea aqueous solution system followed by mixing the resulting solution with a sodium montmorillonite suspension. The suspension then underwent a sol-gel process, solvent replacement, and freeze-drying. The addition of montmorillonite accelerated the sol-gel process in the cellulose solution. Montmorillonite adhered to cellulose substrate surface via hydrogen bonding and was embedded in the pore structure of the cellulose aerogel. As a result, the pore diameter of the aerogel decreased and the specific surface area of the aerogel increased. Furthermore, the addition of montmorillonite increased the compressive modulus and density of the cellulose aerogel and decreased volume shrinkage during the preparation process. However, montmorillonite produced no significant effect on the adsorption performance of the cellulose aerogel, which may have been caused by the low microporous and mesoporous ratios of cellulose aerogel.

## Figures and Tables

**Figure 1 polymers-11-00415-f001:**
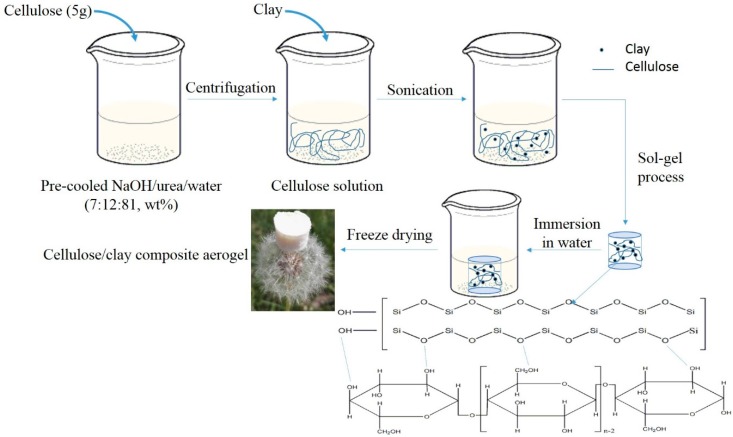
Preparation of a cellulose/montmorillonite composite aerogel.

**Figure 2 polymers-11-00415-f002:**
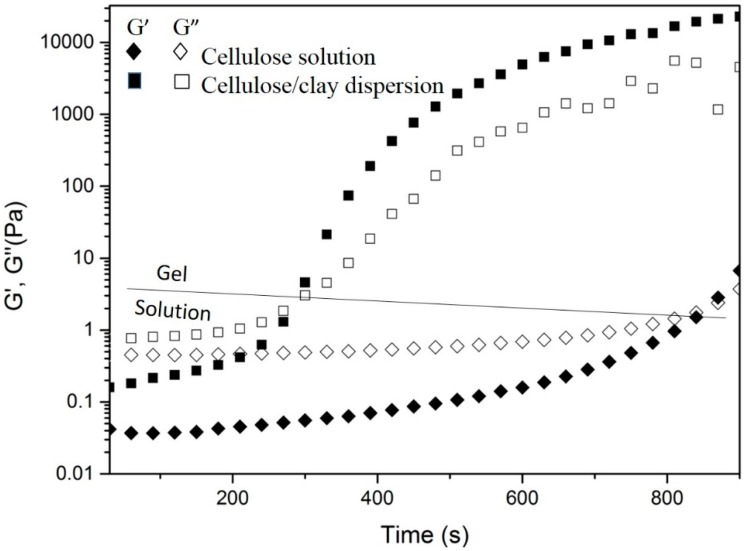
Rheological results of the cellulose solution and cellulose/clay dispersion.

**Figure 3 polymers-11-00415-f003:**
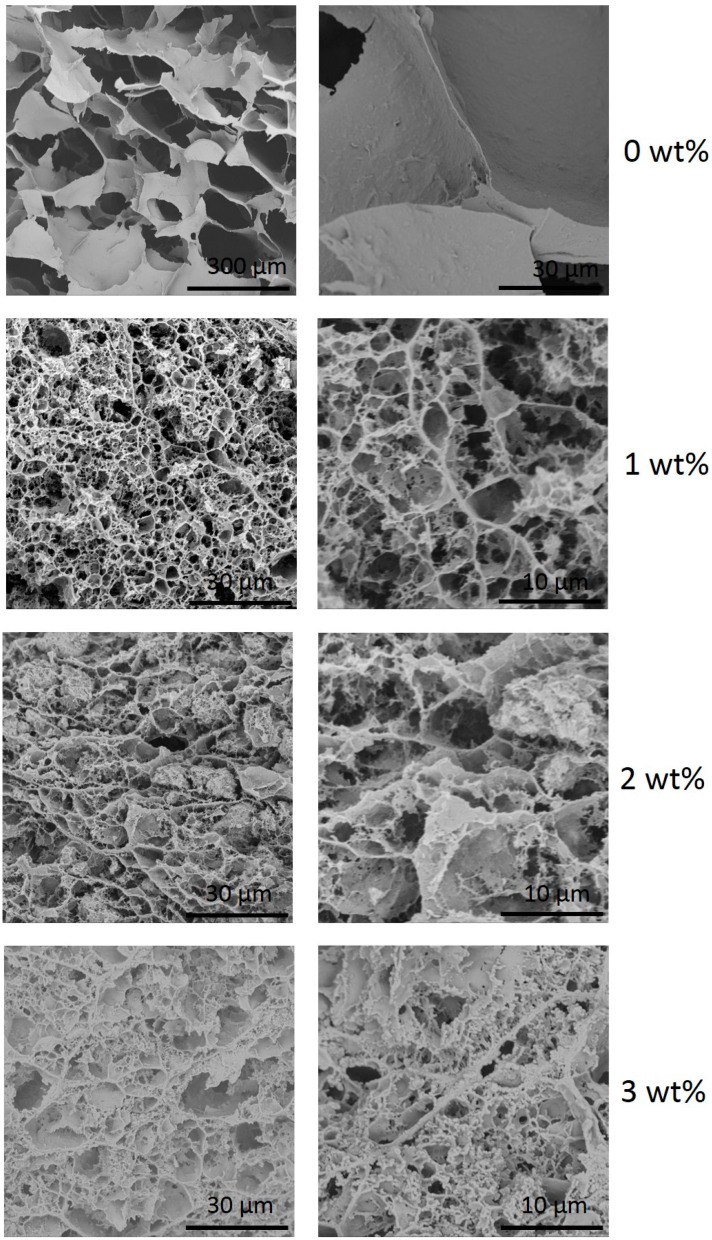
SEM images of cellulose/montmorillonite aerogel with the following montmorillonite ratios: 0, 1, 2 and 3 wt %.

**Figure 4 polymers-11-00415-f004:**
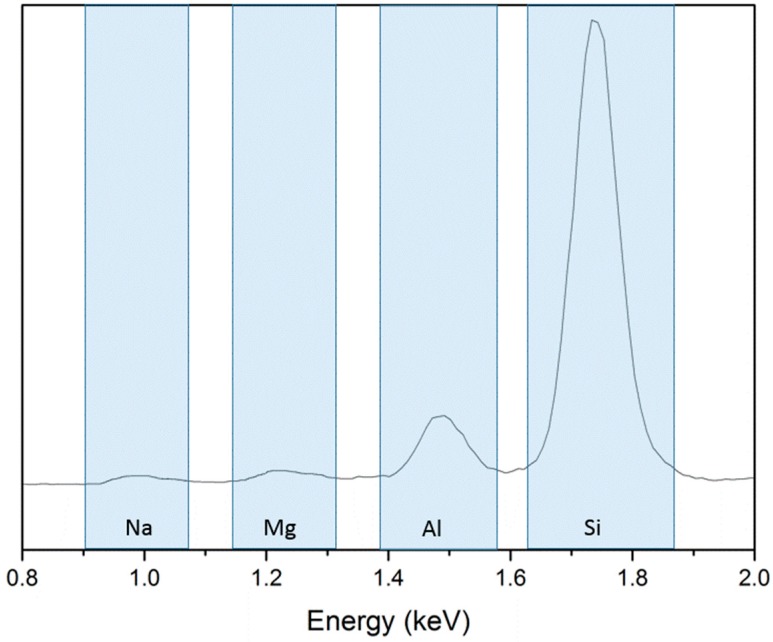
SEM-EDS spectrum of a cellulose–clay aerogel.

**Figure 5 polymers-11-00415-f005:**
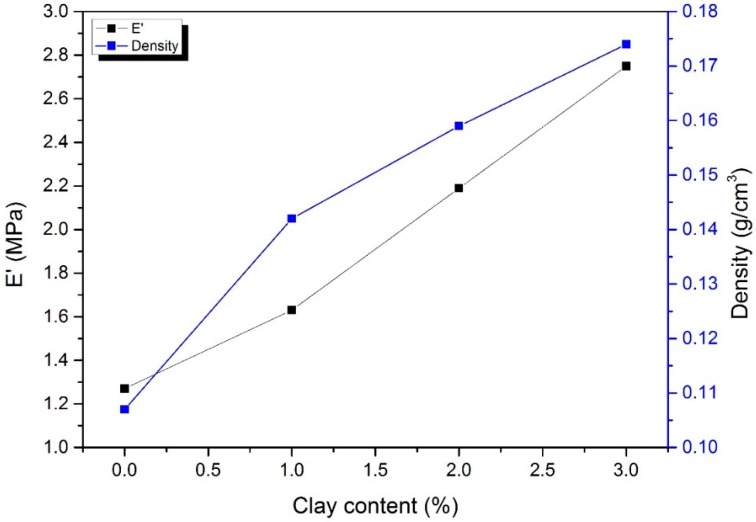
Compression moduli and densities of cellulose aerogels with different montmorillonite concentration.

**Figure 6 polymers-11-00415-f006:**
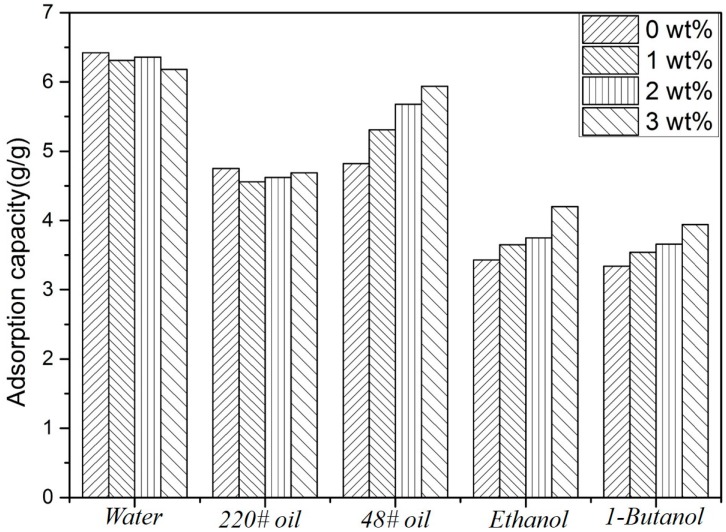
Adsorption capacities of cellulose aerogel and cellulose/clay aerogel.

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
