# Peer review of "Effects of Sodium Montmorillonite on the Preparation and Properties of Cellulose Aerogels"

_polymers, 2019, doi:10.3390/polym11030415_

Round 1

Reviewer 1 Report

The authors do not test any other proportions of cellulose/montmorillonite than 5:3 g. Only after point 3.3 do we know that they tested other mass fractions. This does not help the reader to understand why only 5:3 was thoroughly tested.

The article is not novel and the authors should explain where the novelty is, if there is any.

Author Response

Dear Reviewer 1,

Thank you for your letter and for your comments concerning our manuscript, titled “Effects of sodium montmorillonite on the preparation and properties of cellulose aerogels” (ID: 425148). We found all the comments to be valuable and very helpful for revising and improving our paper as well as the providing additional significance to our research. We have revised the manuscript according to these comments, Reviewer 2’s comments and the referees’ detailed suggestions. The responses to the reviewers are enclosed. We sincerely hope this revised manuscript will be acceptable for publication in Polymers. Thank you very much for all your help. We look forward to hearing from you soon.

Thank you,

Yun-xuan Weng, Ph.D.

Professor

School of Materials and Mechanical Engineering

Beijing Technology and Business University

11 Fucheng Road, Beijing 100048, China

Tel.: +86-10-68985380 Fax.: +86-10-68983573

Yu-zhong Wang

Professor

Center for Degradable and Flame-Retardant Polymeric Materials

College of Chemistry

Sichuan University

Chengdu 610064, China

Tel.: +86-28-85410259

Response to your comments:

Point 1: The authors do not test any other proportions of cellulose/montmorillonite than 5:3 g. Only after point 3.3 do we know that they tested other mass fractions. This does not help the reader to understand why only 5:3 was thoroughly tested.

Response 1: To address this concern, we added information on the effects of the montmorillonite content on the adsorption capacity and pore structure of the fabricated cellulose aerogels.

Point 2: The article is not novel and the authors should explain where the novelty is, if there is any.

Response 2: We introduce montmorillonite into the cellulose solution. The cellulose molecular chain in the solution moves freely in the solvent. After the addition of montmorillonite, the oxygen-containing groups on the sheet interact with the hydroxyl groups on the cellulose molecular chain to form hydrogen bonds. Thereby regulating the microstructure and properties of the aerogel

For the first time, the effects of montmorillonite on the preparation and properties of cellulose aerogels were studied using a rotational rheometer, scanning electron microscope, dynamic thermomechanical instrument and specific surface area. In addition, the liquid adsorption properties of the composite aerogels were studied.

Reviewer 2 Report

Polymers-425148

Title: “Effect of sodium montmorillonite on preparation and properties of cellulose aerogel”

In this paper, the authors use a NaOH/urea aqueous solution system for dissolve cellulose. Besides, they prepare cellulose/montmorillonite aerogel with three-dimensional porous structure. The presence of montmorillonite accelerate the sol-gel process of cellulose solution. Montmorillonite adheres to the surface of cellulose aerogel, reducing the pore diameter of the aerogel and increase its specific surface area, also increase the density and reduces the volume during the preparation process.

The manuscript contains a lot of useful information to readers.

I recommend that the manuscript may be accepted for publication but after major revision.

Comments:

1)      Line 128-130

Comment with major detail the paragraph: “And the initial·····addition of montomorillonite”

2)      Figure 4

The elemental composition is Na, Mg, Al and Si, but in Fig. 4 the Na is not observed

3)      Line 173-175

The formation of hydrogen bonds. The authors can comment with major detail the formation of the bonds Added techniques used.

4)      Revise the English of the manuscript

Other comments:

1)      Line 35

CHANGE      N, N’.methylene bisacrylamide

FOR                N,N’.methylenebisacrylamide

Revise nomenclature of the manuscript

2)      Line 37

Write   90.30%-99.02%          or         90.3%.99.0%3

3)      Line 38

CHANGE      Cu (II)                        FOR    Cu/(II)

4)      Line 47

CHANGE      aerogels which           FOR    aerogels, which

5)      Line 60

Revise the paragraph: “Montmorillonite is a clay which is low-cost”

6)      Line 67

CHANGE      time. And                   FOR    time. The ligand

7)      Line 71

Define the lettes: um (unit mass)

8)      Line 78, 79, 80

CHANGE      3g        FOR    3 g

                        100g    FOR    100 g

                        3g        FOR    3 g

9)      Line 80

400 r/min        Define the units “r”

10)  Line 89, Figure 1

Delete: Figure of the

11)  Line 114

CHANGE      HZ                  FOR    Hz

12)  Line 122

CHANGE      Point               FOR    point

13)  Line 135

CHANGE      fig. 3, fig.4      FOR    Fig.3, Fig. 4

14)  Line 143

CHANGE      Si; And           FOR    Si, and

15)  Line 150

CHANGE      Zhou group     FOR    Zhou et al.

Revise all manuscript

16)  Line 184

Define the letters “PP”

17)  Line 195

CHANGE      we fabricate    FOR    We fabricate

18)  Revise the presentation of the references

[14] [19]

Author Response

Dear Reviewer 2,

Thank you for your letter and for your comments concerning our manuscript, titled “Effects of sodium montmorillonite on the preparation and properties of cellulose aerogels” (ID: 425148). We found all the comments to be valuable and very helpful for revising and improving our paper as well as the providing additional significance to our research. We have revised the manuscript according to these comments, Reviewer 1’s comments and the referees’ detailed suggestions. The responses to the reviewers are enclosed. We sincerely hope this revised manuscript will be acceptable for publication in Polymers. Thank you very much for all your help. We look forward to hearing from you soon.

Thank you,

Yun-xuan Weng, Ph.D.

Professor

School of Materials and Mechanical Engineering

Beijing Technology and Business University

11 Fucheng Road, Beijing 100048, China

Tel.: +86-10-68985380 Fax.: +86-10-68983573

Yu-zhong Wang

Professor

Center for Degradable and Flame-Retardant Polymeric Materials

College of Chemistry

Sichuan University

Chengdu 610064, China

Tel.:+86-28-85410259

Response to your comments:

Point 1: Line 128-130 Comment with major detail the paragraph: “And the initial·····addition of montomorillonite”

Response 1: The addition of a chemical crosslinking agent or materials possessing oxygen-containing groups generally accelerates gelation [9, 21, 25]. In addition, the initial G' and G'' of the cellulose/montmorillonite suspension are larger than those of the cellulose solution, Indicating stronger intermolecular forces and higher cellulose chain entanglement occur in the cellulose/montmorillonite suspension because of the addition of montmorillonite.

Point 2: Figure 4: The elemental composition is Na, Mg, Al and Si, but in Fig. 4 the Na is not observed

Response 2: Thank you for your helpful evaluation and kind suggestion. We have revised Fig. 4 accordingly.

Point 3: Line 173-175 The formation of hydrogen bonds. The authors can comment with major detail the formation of the bonds Added techniques used.

Response 3: We agree with the referee’s helpful suggestion and apologize for neglecting this information. Our speculation was based on previously published papers, including “FT-IR study of montmorillonite–chitosan nanocomposite materials” and “Nanoreinforced bacterial cellulose–montmorillonite composites for biomedical applications”. We have added these two papers to the Reference section.

Point 4:      Revise the English of the manuscript

Response 4: After revising the manuscript ourselves, we used a professional English editing service to further polish our manuscript.

Point 5:      Line 35 CHANGE      N, N’.methylene bisacrylamide FOR                N,N’.methylenebisacrylamide; Revise nomenclature of the manuscript

Response 5: We apologize for the use of incorrect nomenclature. We have revised these instances according to your comment.

Point 6:     Line 37 Write   90.30%-99.02%          or         90.3%.99.0%3 Response 6: We have made this correction according to this comment.

Point 7:      Line 38 CHANGE      Cu (II)                        FOR    Cu/(II)

Response 7: We have made this correction according to this comment.

Point 8:      Line 47 CHANGE      aerogels which           FOR    aerogels, which

Response 8: We have made this correction according to this comment.

Point 9:      Line 60 Revise the paragraph: “Montmorillonite is a clay which is low-cost”

Response 9: We have revised the paragraph as follows, “Sodium-based montmorillonite possesses excellent swelling properties, cation exchangeability, dispersibility in aqueous media, viscosity, lubricity and thermal stability as well as a high hot and wet compressive strength and compressive modulus [17,18].”

Point 10:      Line 67 CHANGE      time. And                   FOR    time. The ligand

Response 10: We have made this correction according to this comment.

Point 11:     Line 71 Define the lettes: um (unit mass)

Response 11: We apologize for this mistake. “µm” is the symbol for “micrometer(s)”.

Point 12:     Line 78, 79, 80

CHANGE      3g        FOR    3 g; 100g    FOR    100 g; 3g        FOR    3 g

Response 12: We have made this correction according to this comment.

Point 13:      Line 80 400 r/min        Define the units “r”

Response 12: We apologize for this mistake. “r” is the symbol for “revolutions”. We have clarified this issue in the text as follows, “4000 revolutions/min”.

Point 14:  Line 89, Figure 1 Delete: Figure of the

Response 14: We have made this correction according to this comment.

Point 15:  Line 114 CHANGE      HZ                  FOR    Hz

Response 15: We have made this correction according to this comment.

Point 16:  Line 122 CHANGE      Point               FOR    point

Response 16: We have made this correction according to this comment.

Point 17:  Line 135 CHANGE      fig. 3, fig.4      FOR    Fig.3, Fig. 4

Response 17: We have made this correction according to this comment.

Point 18:  Line 143 CHANGE      Si; And           FOR    Si, and

Response 18: We have made this correction according to this comment.

Point 19:  Line 150 CHANGE      Zhou group     FOR    Zhou et al.

Revise all manuscript

Response 19: We have made this correction according to this comment.

Point 20:  Line 184 Define the letters “PP”

Response 20: We apologize for this mistake. “PP” is the symbol for “polypropylene”. We have clarified this issue in the text as follows, “commercial adsorbent polypropylene”.

Point 21:  Line 195 CHANGE      we fabricate    FOR    We fabricate

Response 21: We have made this correction according to the Reviewer’s comments.

Point 22:  Revise the presentation of the references [14] [19]

Response 22: We have updated the references as follows:

14.     Liu, X.; Yang, R.; Xu, M.; Ma, C.; Li, W.; Yin, Y. Hydrothermal Synthesis of Cellulose Nanocrystal-Grafted-Acrylic Acid Aerogels with Superabsorbent Properties. Polymers (Basel). 2018, 10, 1168-1183, doi:10.3390/polym10101168.

19.   Erlandsson, J.; Franc, H.; Marais, A.; Granberg, H.; Wågberg, L. Cross-Linked and Shapeable Porous 3D Substrates from Freeze-Linked Cellulose Nanofibrils. Biomacromolecules. 2019, 20, 728-737, doi:10.1021/acs.biomac.8b01412.

Round 2

Reviewer 1 Report

The authors have significantly improved the quality of the manuscript.

Reviewer 2 Report

After the changes, this paper can be accepted, in present form, for their publication in Polymers